# A Multi-Model Diagnosis Method for Slowly Varying Faults of Plunger Pump

Changli Yu [1,2], Haodong Yan [1], Xingming Zhang [1,2,*] and Hua Ye [1]

1 School of Ocean Engineering, Harbin Institute of Technology, Weihai 264209, China
2 Shandong Institute of Shipbuilding Technology, Weihai 264209, China
* Correspondence: zh_xm@hit.edu.cn

**Abstract:** As the energy supply component of hydraulic transmission systems, the plunger pump is widely used in the field of ship and ocean engineering. Thus, its fault diagnosis is of great importance. The multi-model fault diagnosis method based on the Kalman filter is slow in detection and isolation in the process of slowly varying fault diagnosis, and it may be diagnosed as a false failure. In this article, to improve the performance of the multi-model fault diagnosis method, we combine the method and support vector machine and propose a new method by fusing the conditional probability of the multi-model with the posterior probability of the support vector machine. The experimental results on a marine plunger pump illustrate the effectiveness of the proposed method. With the appropriate weight coefficient, the detection speed and isolation speed of the joint multi-model method are improved after the combination of the support vector machine, and the new method has better robustness.

**Keywords:** improved multi-model; probabilistic fusion; fault diagnosis; slow-growing faults

## 1. Introduction

In recent years, fault diagnosis of marine equipment has attracted many researchers. Fault diagnosis methods can be classified into model-based methods, signal-based methods, knowledge-based methods, hybrid methods, and active fault diagnosis methods [1]. Xu et al. [2] proposed a model-based fault detection and isolation scheme for the ship rudder servo system. This algorithm has the potential to perform fault diagnosis autonomously, so it can save time and human resources in sea travel. The experiment shows that four of the six faults can be isolated. The numerical simulation further shows that if the spool displacement sensor is added, all faults can be isolated. Zhong et al. [3] proposed an intelligent fault fusion diagnosis method for marine diesel engines. The combination of D-S evidence theory and GA-SVM algorithm can solve the problem that the diagnosis model based on single sensor data is vulnerable to environmental noise and has low diagnosis accuracy. In addition, this information fusion method can also reduce the risk of overfitting of GA-SVM algorithm and improve its generalization ability. The author claims that the accuracy of the fault diagnosis model based on information fusion can reach 94.17%. Nguyen et al. [4] proposed a new method to process MDIR data of vibration signals. The proposed MB-DNN has a higher classification accuracy and strong noise resistance, which can be used for the early detection of bearing faults. Hoang et al. [5] improved the performance of the motor bearing fault diagnosis method based on the motor current signal by using deep learning and information fusion. Compared with the traditional motor bearing fault diagnosis method based on vibration signal sum, this method has the advantages of simple signal acquisition and lower cost, but poor performance. Maamouri et al. [6] designed a hybrid model-based and signal-based method for fault diagnosis of sensorless speed-controlled induction motor drive (IM) and IGBT open circuit switch. Through various experiments, it is proved that this diagnosis method is efficient, simple, and independent of the transient state and parameters of the motor.

In many cases, researchers often cannot obtain sufficient data about the research object. Therefore, model-based fault diagnosis is necessary. Marine equipment operates in harsh environments, such as those with shock vibrations, high temperatures, and high pressures, which hide the risks of many kinds of failures. The ability to quickly detect and isolate faults is critical to ensuring the stable operation of machinery and equipment, and a multi-model (MM) fault diagnosis method can realize that objective [7]. Under normal circumstances, the state transitions of a system follow certain physical principles, but when a fault occurs, the system state transition process also changes. For all possible situations, MM establishes corresponding state models, and the residuals obtained after filtering with each model are used as the basis for fault diagnosis. A smaller model residual illustrates a closer relationship between the model corresponding to the current situation and the situation itself.

The fault diagnosis method by MM is commonly used in slow-grow fault diagnosis, and researchers have applied this method to many fields. To simultaneously diagnose the gas path and sensor faults, Yang et al. [8] proposed an MM method based on the chi-square test to effectively overcome the problem of misdiagnosis or missed diagnosis caused by fault coupling. The simulation results show that the proposed method has 97% and 94% accuracy in the detection and isolation of sensor fault and gas path fault under a single coupling fault. Zhao et al. [9] proposed an MM method for aeroengine sensor fault diagnosis and estimation. By designing corresponding aeroengine Kalman filter banks and using a hierarchical architecture approach, sensor fault diagnosis can be effectively realized. He et al. [10] proposed a fault diagnosis method for complex chemical process based on MM fusion. The simulation results on the Tennessee Eastman Process dataset and Fluidized Catalytic Cracker fractionation unit dataset show that this method has significant advantages over traditional diagnostic methods in terms of diagnostic precision and recall. Niu et al. [11] studied the problem of fault diagnosis of launch vehicle actuators by using an MM method. The simulation results show that the method can quickly and accurately find rocket fault samples. Sidhu et al. built a state model for a lithium battery [12], and diagnosed overcharge and over-discharge faults based on the conditional probabilities calculated by the filtering residuals. The experiment proved the effectiveness of the MM, but the conditional probability calculated during the diagnosis process fluctuated to a certain extent, which affects the evaluation effect. Pratama et al. applied an MM to fault diagnosis for a differential drive robot [13], and the MM could accurately evaluate the states of the left and right rotors of the robot. Naderi et al. established a set of state models to diagnose the failures of a gas turbine [14], such as turbine efficiency drops and compressor efficiency drops, evaluated the current state by the corresponding conditional probability, and proved that the selected unscented Kalman (UKF) had faster detection and isolation speeds than extended Kalman (EKF).

In recent years, many scholars have improved the MM method. To mitigate the problem that the number of state models that need to be established increases exponentially when the number of failures increases, Gao et al. believed that multiple faults occurred one after another and had a buffer time [15]; thus, he proposed a satellite-like hierarchical structure, and the filter bank corresponding to the fault model was activated after a fault was isolated. Sadeghzadeh et al. used graph theory to decouple a robot navigation system into multiple subsystems and established state models for each subsystem [16]. Experiments showed that this method not only reduces the number of models but also effectively isolates sensor faults, such as inertial sensors and camera sensors. To improve the problem that the diagnosis effect may be reduced due to insufficient state model accuracy, Zhu et al. used a kernel function instead of a state transition function to establish the state model [17], and this method can detect typical actuator faults in real time. Yang et al. used a strong tracking EKF for filtering to ensure that the filtering residual obeyed a Gaussian distribution [18], improving the robustness of the MM by reducing the speed of fault detection and isolation. Zhao et al. presented a method for estimating the severity of faults [19]; experiments

showed that as the degree of fault severity decreased, the required detection and isolation speeds also increased.

During the evolution processes of slowly changing faults, early fault diagnosis plays an important role in ensuring the stable operation of mechanical equipment, so it is of great significance to improve the detection and isolation speeds of MM fault diagnosis methods. However, few of the above methods improve the speed of MM diagnosing faults. In the early stage of fault evolution, due to the insignificant fluctuations of parameters or the characteristics of other faults, the speed and robustness of MM designed for fault diagnosis are reduced. Therefore, this article uses a support vector machine (SVM) to improve the performance of the MM method by fusing the posterior probability of the SVM and the conditional probability of the MM, thereby improving the detection and isolation speeds of the MM and enhancing its robustness. Finally, experiments with a marine plunger pump are carried out to verify the algorithm.

## 2. SVM-MM Algorithm Framework

An MM calculates a conditional probability based on the residual of the state model, which takes measurement data as input. The conditional probability can be regarded as the occurrence probability of the state corresponding to the model. However, due to the state fluctuations of the system in the fault evolution stage, and the corresponding residuals are insignificant, the value of conditional probability increases delay, which affects the diagnosis speed of the MM. In this article, by exploiting the sensitivity of an SVM to data, the posterior probability of the SVM's classification result is fused with the conditional probability of an MM, and fault diagnosis is performed according to the fused probability.

The difference between SVM-MM and the traditional multiple models lies in the calculation process of conditional probability. SVM-MM enhances MM's ability to evaluate the current state by fusing the posterior probability of SVM. First, determine the potential operating state of the machine, and establish corresponding state models to form filter banks. Then, the currently measured data is used as the filter input, and the residual is calculated according to Equation (34). The conditional probability of MM is calculated by using the residual of each state model through Equation (5). The residual of the normal state model is used as the SVM input so that it can classify the current state and calculate the corresponding posterior probability according to Equation (11). Then, a new conditional probability is formed by fusing the posterior probability of SVM and the conditional probability of MM according to Equation (12). Finally, fault diagnosis and isolation are realized through Equation (13) according to the updated conditional probability and threshold, as shown in Figure 1.

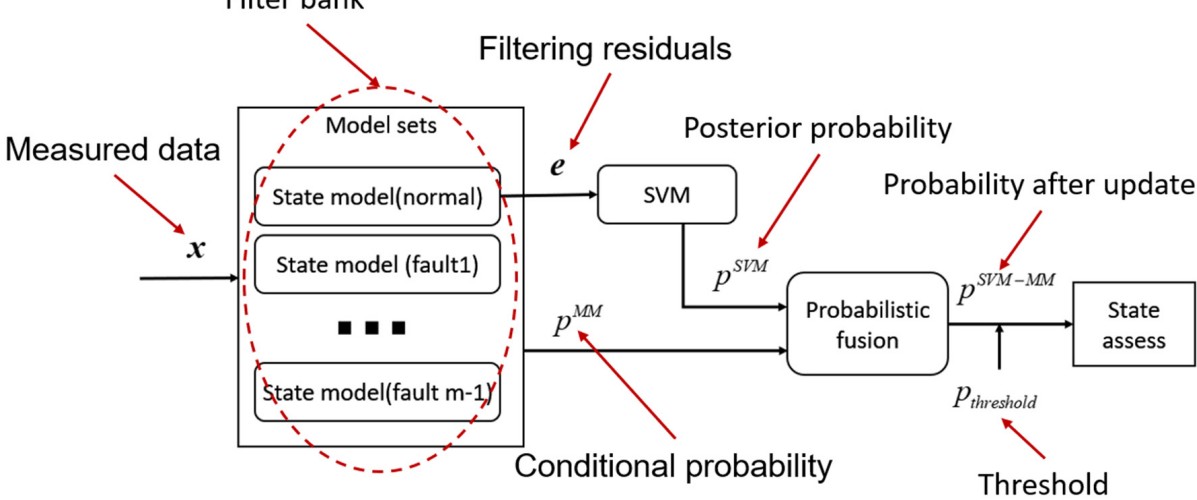

**Figure 1.** Architecture of the SVM-MM.

### 2.1. State Model

The dynamic characteristics of a mechanical system can be comprehensively and accurately described by using the parameter set $x_k$ with the smallest number of parameters at any time, and $x_k$ is called the state variable of the system. The relationship between this state variable and the state variable at the previous moment can be described as:

$$x_k = f(x_{k-1}, u_{k-1}) + w_k \tag{1}$$

where $f$ is the system state transition equation; $u$ is the control input variable, which is the set of parameters that affect the state of the system; $w$ is the system noise, including the error of the system state transition equation; $k - 1$ represents the time when $t = k - 1$.

The output variable **y** of the system is generally selected as the set of parameters that can be observed by the system, and the relationship between it and state variable $x$ at time $k$ can be described as:

$$y_k = g(x_k, u_k) + v_k \tag{2}$$

where $g$ is the measurement conversion equation; $v_k$ is the measurement noise, such as that from the sensor error; $k$ represents the time when $t = k$.

The state model of the system can be described as a combination of Equations (1) and (2).

### 2.2. Conditional Probability of the MM

Suppose that the number of possible operating states for the mechanical system $S_i(i = 0, 1, \ldots m - 1)$ is $m$, and regard the mechanical system as a first-order Markov Chain; then, the time $k$ is only related to the previous time $k - 1$. Then, the running state of the plunger pump can be represented by Markov chain, as shown in Figure 2. Combining the data measured at time $k$ and the previous time, the conditional probability of being in state $S_j$ at this time can be described as:

$$p_{j,k} = p(S_j | J_k) \tag{3}$$

where $p_{j,k}$ is the conditional probability of being in state $S_j$ at time $k$; $S_j$ is the $jth$ state with $1 \leq j \leq m$; $J_k$ is the set of historical measurement data at time $k$ with $J_k = (y_0, y_1, y_2, \ldots y_k)$.

According to the residuals and covariances of the outputs of each model, the conditional probability density corresponding to each state at time $k$ can be described as:

$$f(y_k | S_i, J_k) = \frac{exp\left(-\frac{1}{2} e_{i,k}^T P_{zz,i,k}^{-1} e_{i,k}\right)}{(2\pi)^{\frac{n}{2}} |P_{zz,i,k}|^{\frac{1}{2}}} \tag{4}$$

where $f(y_k | S_i, J_k)$ is the conditional probability density of being in state $S_j$; $e_{i,k}$ and $P_{zz,i,k}$ are the residuals and covariances of the output variables, respectively (refer to the Appendix A for the specific solution steps); and $n$ is the dimensionality of the measurement data.

Combining Equations (3) and (4). according to Bayes' theorem, the conditional probability corresponding to each state at time $k$ is rewritten as:

$$p_{i,k}^{MM} = \frac{f(y_k | S_i, J_k) p_{i,k-1}}{\sum_{j=0}^{m-1} f(y_k | S_j, J_k) p_{j,k-1}} \tag{5}$$

where $p_{i,k}^{MM}$ is the conditional probability of being in state $S_j$ at time $k$ calculated by an MM. Considering that the mechanical system may undergo state transitions during operation, to ensure that the MM can be responded to in time, a minimum value is set for each probability $(minp^{MM} = 0.01)$.

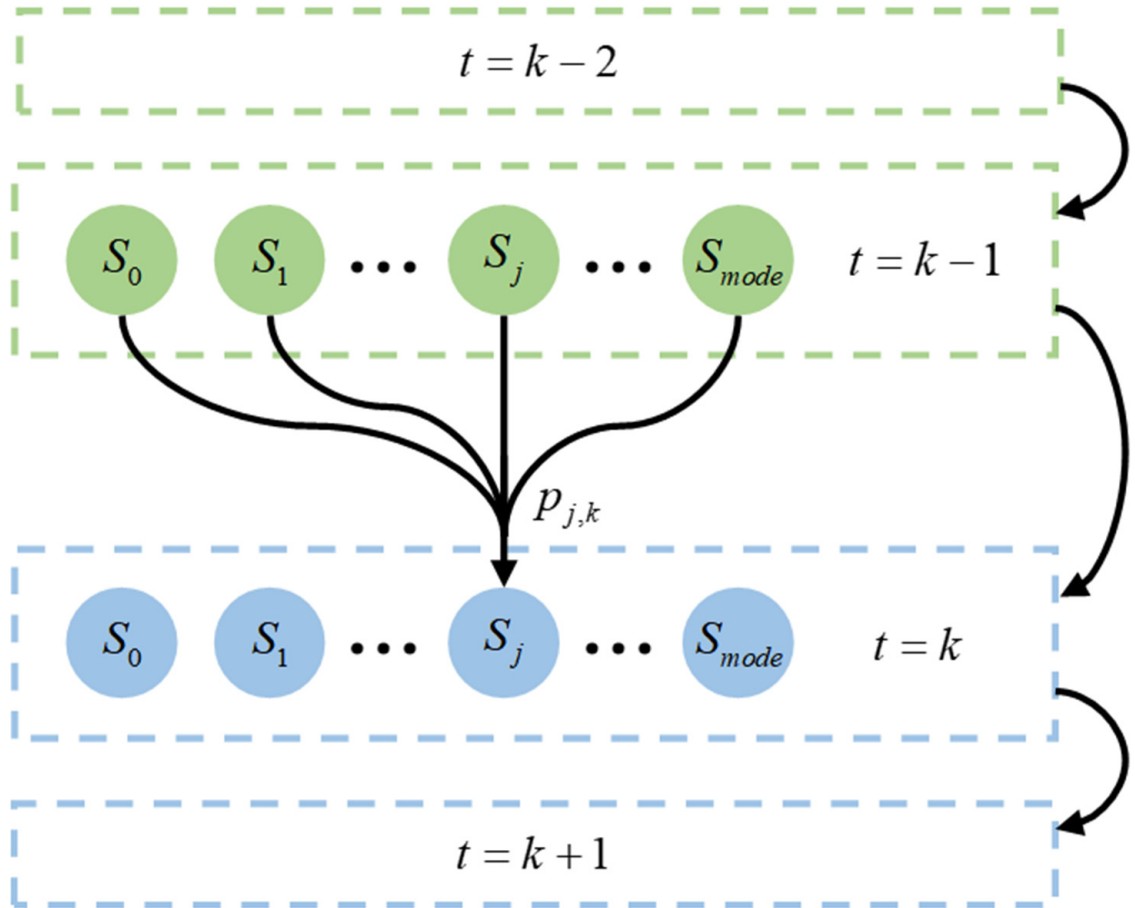

**Figure 2.** Markov Chain.

The filtering residuals of each state model can be obtained from the set of state models established in advance with the currently measured data as input and then the conditional probability can be calculated according to Equation (5), which reflects the probability of the corresponding state. Because of the characteristics of the Bayesian theorem, the occurrence probability of these states is only related to the previous time, and at the same time, the sum of conditional probabilities of all models in the state model group is 1.

*2.3. Posterior Probability of the SVM*

The principle of the SVM is to find a hyperplane in the set space formed by multiple sets of data:

$$h(\boldsymbol{a}) = \boldsymbol{w} \cdot \varphi(\boldsymbol{a}) + c = \sum_{i=1}^{n} \boldsymbol{w}_i \cdot \varphi(\boldsymbol{a}) + c = 0 \tag{6}$$

where $\boldsymbol{w}$ is the weight vector, $h$ and $\varphi$ are hyperplane functions and sample functions, respectively; $\boldsymbol{a}$ denotes the sample data, and $c$ is the threshold.

The goal is to satisfy the following optimization problem:

$$min\frac{1}{2}\boldsymbol{w}^T\boldsymbol{w} + C\sum_{i=1}^{n}\xi_i s.t.b_i(\boldsymbol{w}\cdot\varphi(\boldsymbol{a}_i)+c) \geq 1-\xi_i, \xi > 0 \tag{7}$$

where $C$ is the penalty function; $\xi$ is the slack variable; $b$ is the predictor for the sample.

Refer to Lagrange dual functions, the optimization problem can be described as:

$$\begin{cases} max \sum_{i=1}^{n} \alpha_i - \frac{1}{2} \sum_{i=1}^{n} \sum_{j=1}^{n} \alpha_i \alpha_j b_i b_j \kappa \left( \boldsymbol{a}_i, \boldsymbol{a}_j \right) \\ s.t. \sum_{i=1}^{n} \alpha_i b_i = 0, \alpha_i \geq 0, i = 1, 2 \ldots n \end{cases} \tag{8}$$

where $\alpha$ is the Lagrange coefficient and $\kappa$ is the kernel function, which is selected as the Gaussian kernel function in this article.

Let $\alpha^*$ be the solution of Equation (8); then, the solutions of the threshold $c^*$ and weight vector $\boldsymbol{w}^*$ can be described as:

$$\begin{cases} c^* = y_l - \sum_{i=1}^{n} \alpha_i^* y_i \kappa (\boldsymbol{a}_i, \boldsymbol{a}_l) \\ \boldsymbol{w}^* = \sum_{i=1}^{n} \alpha_i^* y_i \kappa (\boldsymbol{a}_i, \boldsymbol{a}_l) \end{cases} \tag{9}$$

Substituting Equation (9) into Equation (8), the discriminant classification function of the SVM can be described as:

$$h(\mathbf{a}) = sgn \left[ \sum_{i=1}^{n} \alpha_i^* y_i \kappa (\mathbf{a}_i, \mathbf{a}_l) + c^* \right] \tag{10}$$

During the process of fault diagnosis, the actual measurement data includes noise and outliers; the Kalman filter can reduce the influence of noise; and the characteristics of faults can be represented by the residuals from the normal state model, thus the residuals can be regarded as sample data with denoising and feature extraction. In this article, we use the residuals from the normal state model to train SVM, the posterior probability that the SVM classifies the current state as $S_i$ can be described as [19]:

$$p_{i,k}^{SVM} = p(b = i \,|\mathbf{a}) = \frac{1}{1 + exp(A_i h(\mathbf{a}) + B_i)} \tag{11}$$

where $p_{i,k}^{SVM}$ is the posterior probability of classifying the input as being in state $S_j$ at time $k$ calculated by the SVM; $\boldsymbol{a}$ and $b$ are the input and output of the SVM, respectively; $A$ and $B$ are the coefficients yielded by fitting and training.

*2.4. Probabilistic Fusion*

The information from different sources can be integrated by probabilistic fusion, and the fused probability has the completeness and consistency of mathematics more confidently [20]. In this article, the probability distribution of $p^{MM}$ and $p^{SVM}$ are the same, so we can fusion the two probabilistic by referring to Bayesian melding, as shown in Figure 3.

The common fusion methods are linear and logarithmic, we choose linear fusion [21]:

$$p_{i,k}^{SVM-MM} = \alpha_p p_{i,k}^{MM} + \left( 1 - \alpha_p \right) p_{i,k}^{SVM} \tag{12}$$

where $\alpha_p$ is the weight factor for the conditional probabilities calculated by the MM. Due to this probability is related to historical measurement data, it has a certain robustness to abnormal inputs, while the posterior probability calculated by the SVM is only related to the current moment, so it should have a larger weight.

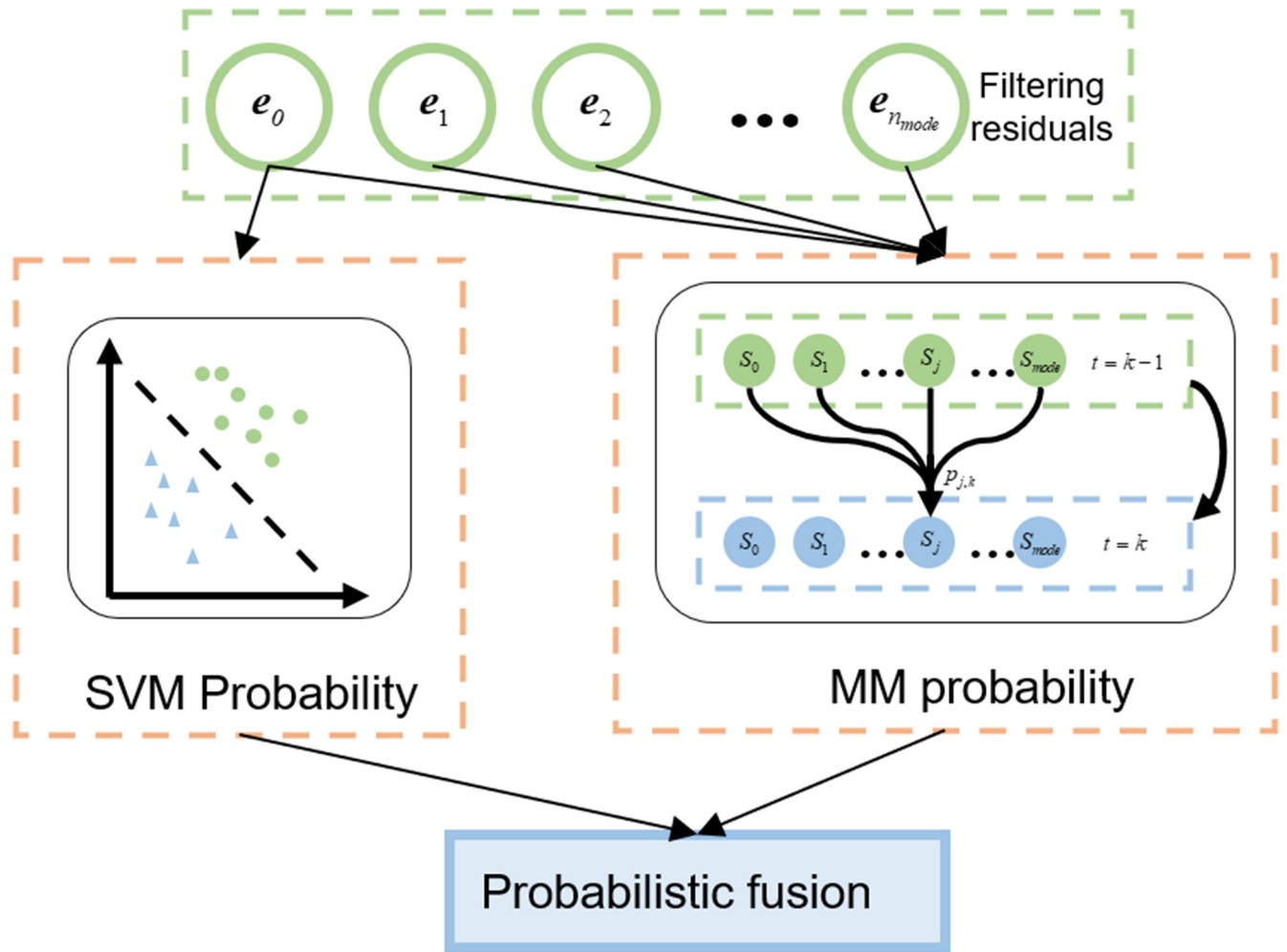

**Figure 3.** Process of probability fusion.

*2.5. State Assessment*

According to the conditional probability obtained after fusion, the current state can be evaluated in combination with the threshold:

$$S_{i,k} = arg \left| p_{i,k} > p_{threshold}, i \in [0, m-1] \right| \tag{13}$$

where $S_{i,k}$ means that the system is in state $S_i$ at time $k$; $p_{threshold}$ is a threshold. If the calculated probability exceeds the threshold, the corresponding state is considered to exist, and if the probability is lower than the threshold, the corresponding state is considered to not exist. The correction effect of SVM is inversely proportional to the threshold value. If the value is too small, the instability of SVM will be improved. If the value is too large, the correction effect is not obvious. In this article, $p_{threshold} = 0.95$.

## 3. Research Object

The plunger pump is the energy supply component of the hydraulic system of offshore equipment, which is widely used in many fields, such as offshore power plants, offshore oil and gas platforms, offshore buoys, and underwater vehicles. Therefore, this paper takes the piston pump as the research object to verify the effectiveness of the fault diagnosis method.

*3.1. The Plunger Pump Mechanism Model*

Currently, machinery and equipment are complex and highly nonlinear systems, and the types of possible failures vary. To verify the effectiveness of the method proposed in this

article, the selected research object is a swash plate axial piston pump. The plunger pump works in a location with large temperature changes and severe vibrations and shocks. Its main parameters are shown in Table 1, and the structural principles are shown in Figure 4. The plunger pump stroke is related to the distribution circle radius of the plunger $R$ and the inclination of the swash plate $\gamma$:

$$S = 2Rtan\gamma \tag{14}$$

**Table 1.** Parameters of the Plunger Pump.

| Parameter | Value |
|---|---|
| rotation speed | 5000 r/min |
| outlet pressure rating | 16 MPa |
| oil return pressure | 0.3~0.55 MPa |
| inlet pressure | 0.25~0.3 MPa |
| egress traffic | 7 L/min |
| return oil flow | <0.4 L/min |

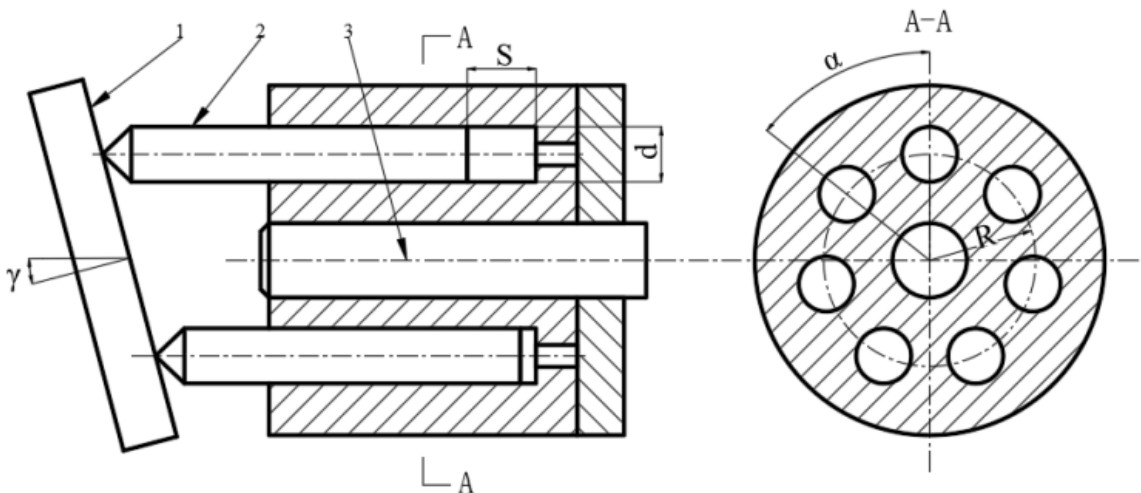

**Figure 4.** Plunger pump structure (1: swash plate; 2: plunger; 3: pump shaft).

A single plunger completes two actions (oil absorption and oil discharging) every time it goes back and forth, and the plunger pump completes one plunger's oil absorption and oil discharge pair after one revolution. Therefore, the average displacement of the pump $q$ is related to the number of plungers $z$ and the plunger pump speed $N$:

$$q = zNV = \frac{\pi d^2}{2} zNRtan\gamma \tag{15}$$

where $V$ is the maximum volume change of a single plunger cavity and $d$ is the plunger diameter.

The torque $M_i$ produced by a single plunger on the motor shaft can be described as:

$$M_i = F_i Rtan\gamma sin[\theta + (i-1)\alpha] \tag{16}$$

where $\theta$ is the angle between the centre of a plunger and the vertical plane; $\alpha$ is the angle between two adjacent piston pumps with $\alpha = 2\pi/z$; $F_i$ is the axial pressure.

Then, the relationship between the motor torque $M_\theta$ and $\theta$ can be described as:

$$M_\theta = \sum_{i=1}^{z} M_i = \sum_{i=1}^{z} F_i Rtan\gamma sin[\theta + (z-1)\alpha] \tag{17}$$

In this pump, $z$ is an odd number, and considering the oil suction and discharge stages of the plunger, the above formula is rewritten as:

$$M_\beta = \begin{cases} \frac{\pi d^2(p_o-p_i)Rtan\gamma cos(\theta-\alpha/4)}{8sin(\alpha/4)}, 0 \le \theta \le \frac{\alpha}{2} \\ \frac{\pi d^2(p_o-p_i)Rtan\gamma cos(\theta-3\alpha/4)}{8sin(\alpha/4)}, \frac{\alpha}{2} \le \theta \le \alpha \end{cases} \tag{18}$$

where $p_i$ and $p_o$ are inlet pressure and oil discharge pressure, respectively.

Integrating this formula and calculating its average value, the average torque $M$ can be described as:

$$M = \frac{\pi d^2(p_o - p_i)Rtan\gamma}{2\alpha} \tag{19}$$

*3.2. Fault Simulation and State Model Establishment*

The plunger pump rotates the swash plate by a motor, thereby driving the piston to reciprocate and realize the function of the oil suction pump. Many potential faults can occur, such as improper assembly or motor displacement faults leading to pump shaft tilt, which changes the inclination of the swash plate $\gamma$; oil corrosion faults causing pipeline leakage, which reduces the oil discharge pressure $p_o$; poor plunger lubrication faults causing sticking, which reduces the number of the effective number of plungers $z$; reductions in the spring force caused by fatigue, which reduces the stroke of the piston $S$. In this article, the above types of faults are simulated and used as the data source for experimental verification. The specific scheme is shown in Table 2. Simulated faults vary linearly with time steps, including three stages: normal, fault evolution, and fault. This article simulates the occurrence of various slowly changing faults from Table 2. First, the plunger pump model runs 100 steps under normal conditions; then, fault evolution is performed to change the value within 200 steps, and finally, the model runs 100 steps in the fault state, for a total of 400 steps.

**Table 2.** Fault Simulation Scheme.

| Fault Type | Symbol | Parameter Change |
|---|---|---|
| shaft displacement | $S_1$ | swash plate inclination $\gamma - 2$ |
| pipeline leakage | $S_2$ | discharge pressure $p_o - 2$ |
| piston sticking | $S_3$ | number of plungers $z - 1$ |
| spring breakage | $S_4$ | piston stroke $S - 0.02$ |

According to the mechanism model of the plunger pump and the fault simulation scheme, the corresponding state model is established. Equations (1) and (2) can be rewritten as:

$$x_k = x_{k-1} \tag{20}$$

$$y_k = g(x) = \begin{cases} q = \frac{\pi d^2(z+\Delta z)N2R(tan(\gamma+\Delta\gamma)+\Delta S)}{4} \\ p_o = \frac{2\alpha}{\pi d^2 Rtan(\gamma+\Delta\gamma)} + p_i + \Delta p_o \end{cases} \tag{21}$$

where: $y_k = (N, M, p_i, p_o, q)$, $x_k = (N, M, p_i)$.

**4. Simulation Results**

The purpose of this article is to improve the speed of detection and isolation for an MM. Therefore, in the process of fault diagnosis, the time consumption of fault detection and isolation can be used as a reference for evaluating the improvement effect of the new method [8]:

$$\varepsilon_d = \Delta t_{\text{detection}} / \Delta t_{transition} \tag{22}$$

$$\varepsilon_i = \Delta t_{isolation} / \Delta t_{transition} \tag{23}$$

where $\Delta t_{detection}$ is the time from when a failure begins to occur until the algorithm detects that the system is not in a normal state; $\Delta t_{isolation}$ is the time from when the fault begins to occur until the algorithm correctly isolates the fault; $\Delta t_{transition}$ is the time from when the fault evolves to a steady state; $\varepsilon_d$ and $\varepsilon_i$ are performance indicators that reflect the speed of fault detection or isolation achieved by the method.

*4.1. Single Fault*

This article simulates the occurrence of various slowly changing faults from Table 2. First, the plunger pump model runs 100 steps under normal conditions; then, fault evolution is performed to change the value within 200 steps, and finally, the model runs 100 steps in the fault state, for a total of 400 steps. Since the data is generated by mathematical model simulation, in order to simulate the real measurement, we randomly add 1% Gaussian noise to the data to simulate the sensor measurement. The indicators of detection and isolation that diagnose a single fault are shown in Table 3.

**Table 3.** Single-Failure Diagnosis Performance Indicators.

| Fault Type | Method | $\varepsilon_d$ | $\varepsilon_i$ |
|---|---|---|---|
| | MM | 0.545 | 0.625 |
| $S_1$ | SVM-MM1 | 0.345 | 0.585 |
| | SVM-MM2 | 0.010 | 0.565 |
| | MM | 0.790 | 1.14 |
| $S_2$ | SVM-MM1 | 0.265 | 0.790 |
| | SVM-MM2 | 0.220 | 0.635 |
| | MM | 0.565 | 0.645 |
| $S_3$ | SVM-MM1 | 0.420 | 0.610 |
| | SVM-MM2 | 0.010 | / |
| | MM | 0.430 | 1.135 |
| $S_4$ | SVM-MM1 | 0.325 | 1.025 |
| | SVM-MM2 | 0.009 | 0.995 |

It can be seen from Table 3 that the detection and isolation speeds achieved for the four types of faults are improved by fusing the MM with the posterior probability of the SVM, and the speeds are greatly improved with the reduction of the weight coefficient. During the process of diagnosing fault $S_1$, as shown in Figure 5a, for all steps < 209, the conditional probability $p_0^{MM}$ is not lower than the threshold, which illustrates that the plunger pump is in the normal operation state before step 209, and after step 209, the occurrence of a fault is detected by the MM. At the same time, the $p_1^{MM}$ calculated by the MM in step 225 exceeds 0.95, and fault $S_1$ is isolated. Due to the correction effect of the fusion probability, in the SVM-MM method, the conditional probability of $S_0$ ($p_0^{SVM-MM}$) in step 169 is lower than 0.95, and the fault is detected 40 steps ahead of the point at which it is detected by the MM, and fault $S_1$ is isolated in step 217. The speeds of detection and isolation are 36.70% and 6.4% higher than those of the MM, and the speed improvements obtained with $\alpha_p = 0.95$ are even higher (98.17% and 9.6% higher in terms of detection and isolation). During the processes of diagnosing faults $S_2$ and $S_3$ (Figure 5b,c), the SVM-MM also has faster detection and isolation speeds, and the degree of improvement depends on the value of the weight factor. However, with a weight factor that is too low, the influence of the posterior probability of the SVM increases. As shown in Figure 5c, although the detection speed with $\alpha_p = 0.95$ is faster than that achieved with $\alpha_p = 0.98$, when $\alpha_P = 0.95$ the fluctuation of the case near the threshold of 0.95 affects the evaluation results for the current state. Due to the posterior probability of SVM not being large, the conditional probability after fusion is low and fluctuates. Therefore, the value of the weight for the conditional probabilities directly affects the diagnosis results. The larger the value is, the more the detection and isolation speeds are improved, but the influence of uncertainty of the SVM is larger. The values should be selected based on the performance of the SVM on the training data.

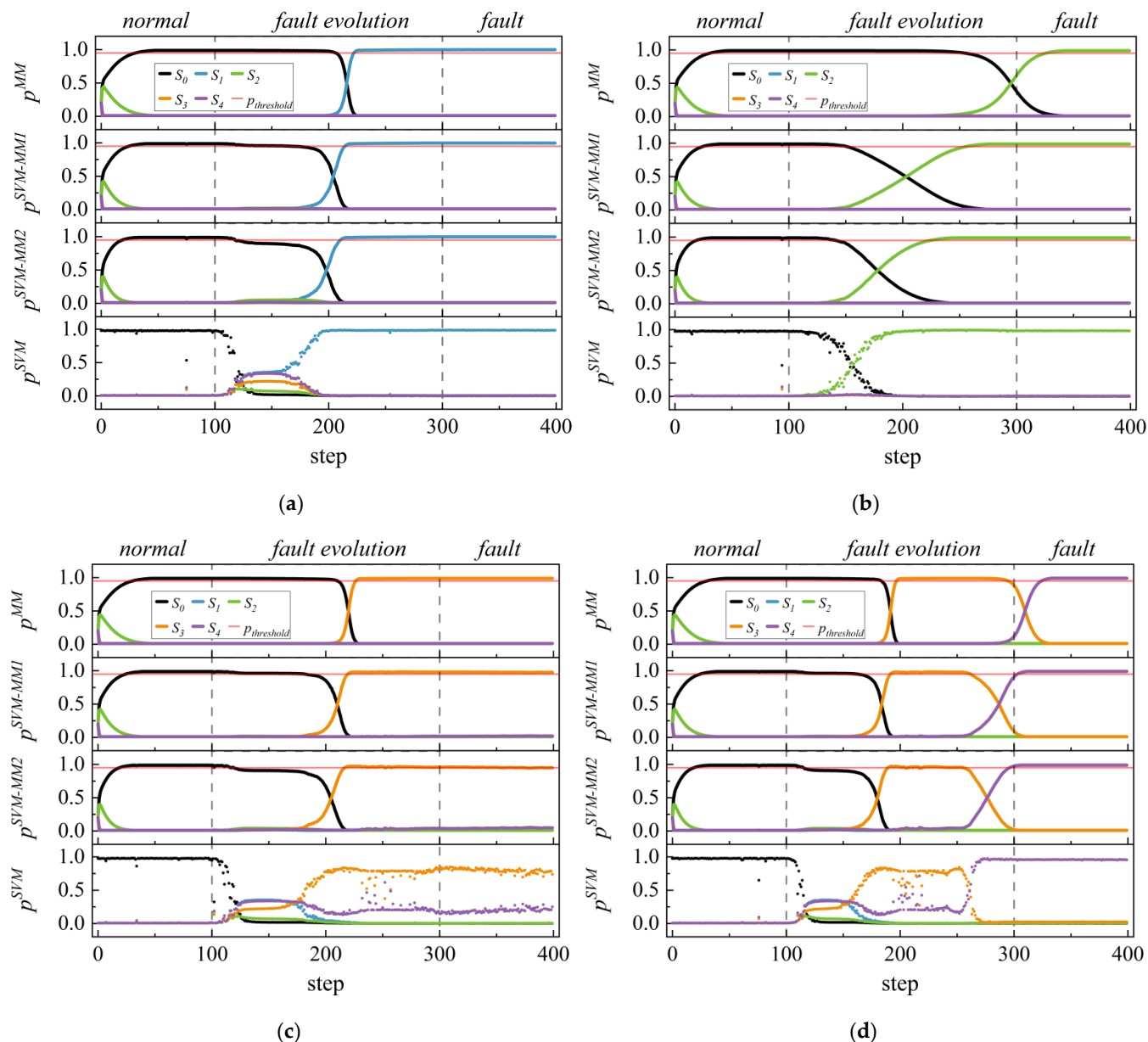

**Figure 5.** Probability calculated by four methods (MM, SVM, SVM-MM1, and SVM-MM2, $\alpha_p$ in SVM-MM1 is 0.98, and $\alpha_p$ in SVM-MM2 is 0.95) during fault diagnosis: (**a**) Fault $S_1$ diagnosis results; (**b**) Fault $S_2$ diagnosis results; (**c**) Fault $S_3$ diagnosis results; (**d**) Fault $S_4$ diagnosis results.

In fault $S_4$ (Figure 5d) the conditional probability of fault $S_3$ calculated by the MM in steps 199 to 297 exceeds the threshold of 0.95, resulting in a false alarm phenomenon; the SVM-MM produces error responses between steps 194 to 264 and 192 to 257, but the false alarm duration is reduced compared with that of the MM, which indicates that the SVM-MM has better robustness.

### 4.2. Multiple Faults

Multiple faults are also concerns in the field of MM diagnosis. In this article, multiple faults have an occurrence sequence. After the first fault is isolated, the state model group is updated to make it capable of diagnosing subsequent faults. As shown in Figure 6, assuming that the occurrence of fault $S_1$ is detected, the state model sets are updated to $S_1$, $S_{12}$, $S_{13}$ and $S_{14}$, and $S_1$ is regarded as a normal state model. Its filter residual is used as the

input of $SVM_1$ to classify and calculate the posterior probability, where $SVM_1$ is the SVM trained after removing the samples of fault $S_1$.

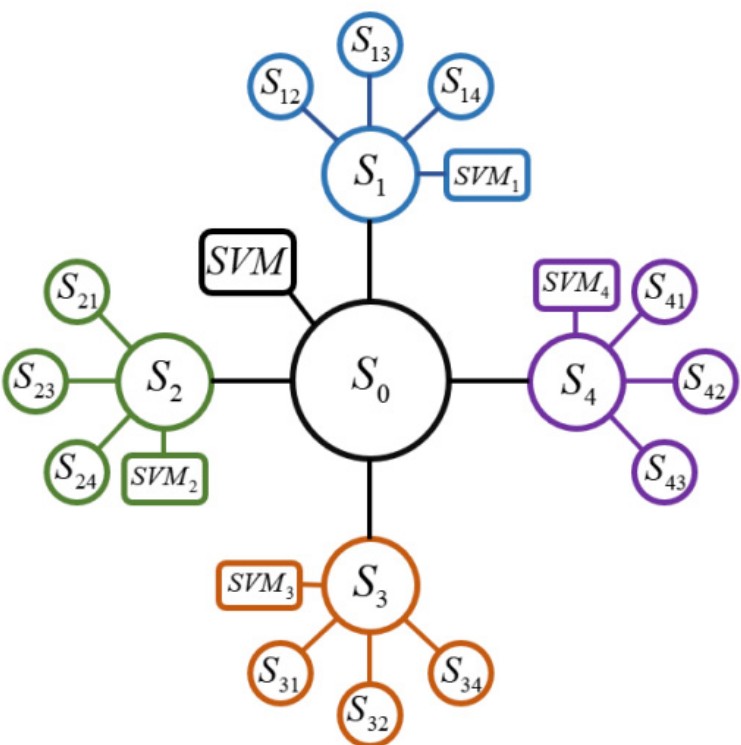

**Figure 6.** SVM-MM framework for multiple-fault diagnosis.

We choose $S_1$ and $S_2$, $S_3$ and $S_4$ as the types of simulation of multiple-fault. The multiple-fault simulation process in this article makes the plunger pump run for 100 steps under normal conditions, evolve to the first fault within 200 steps, then run for 100 steps under the first fault condition, evolve to the second fault within 200 steps, and finally run for 100 steps with both faults present, for a total of 700 steps. Simulated faults vary linearly with time steps. The MM and SVM-MM methods are used for diagnosis, and $\alpha_p = 0.98$ is selected according to the analysis results in the previous section. The indicators of detection and isolation are shown in Table 4. The diagnosis results are shown in Figures 7 and 8.

It can be seen from Table 4 that the SVM-MM can diagnose multiple faults by updating model sets and the SVM. In Figure 7, the SVM-MM detects the occurrence of the second fault 102 steps earlier than the MM and isolates the fault 55 steps earlier, improving the detection and isolation speeds for the second fault by 67.11% and 25.94%, respectively. In Figure 8, the SVM-MM improves the detection and isolation speeds by 20.00% and 4.67% over those of the MM, respectively, indicating that the SVM-MM combined with the SVM posterior probability can also improve the fault diagnosis speed during the process of diagnosing multiple faults.

**Table 4.** Multiple-Fault Diagnosis Performance Indicators.

| Fault Type | Method | First Fault | | Second Fault | |
|---|---|---|---|---|---|
| | | $\varepsilon_d$ | $\varepsilon_i$ | $\varepsilon_d$ | $\varepsilon_i$ |
| $S_1 + S_2$ | MM | 0.545 | 0.620 | 0.760 | 1.060 |
| | SVM-MM | 0.380 | 0.580 | 0.250 | 0.785 |
| $S_3 + S_4$ | MM | 0.560 | 0.645 | 0.475 | 0.535 |
| | SVM-MM | 0.425 | 0.610 | 0.380 | 0.510 |

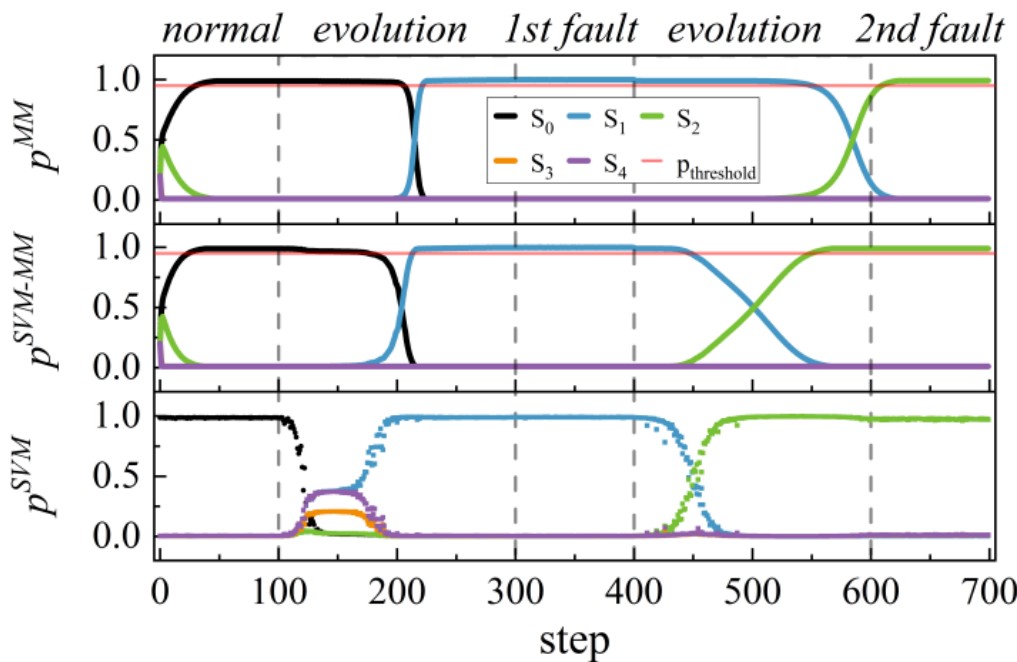

**Figure 7.** Diagnosis results for multiple faults: $S_1$ and $S_2$.

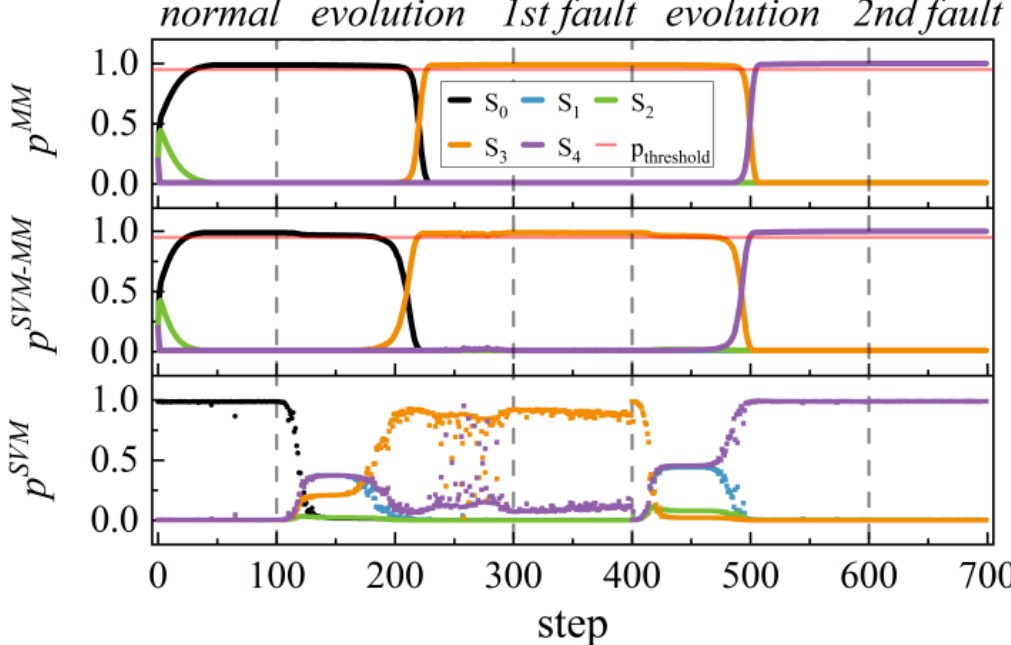

**Figure 8.** Diagnosis results for multiple faults: $S_3$ and $S_4$.

## 5. Conclusions

To mitigate the problem that MM has low detection and isolation speeds and unstable performance during fault evolutions, this article proposes a joint diagnosis method involving an MM and an SVM. The new method integrates the conditional probability of the MM and the posterior probability of the SVM and corrects the conditional probability by evaluating the current operating status of the SVM. By establishing the mechanism model of the plunger pump, several faults, including input load fluctuations, wear, and corrosion are diagnosed. Experiments show that during the process of diagnosing single and multiple fault types, the SVM-MM has faster detection and isolation speeds than the MM. Additionally, the value of the weight factor is discussed. The larger the value is, the

faster the speed, but the influence of the SVM uncertainty is greater. The SVM-MM is also robust to the influences of SVM misclassifications and parameter fluctuations caused by fault evolutions. With the constructed potential fault state model group, the SVM-MM can detect and isolate slowly changing faults for mechanical equipment. For unknown fault diagnosis cases, the method updates the conditional probability in combination with the posterior probability of the SVM, preventing all probabilities from exceeding the threshold so that the model can indicate that the system is currently in a new unknown state, thereby enabling the diagnosis of unknown faults.

**Author Contributions:** Methodology, C.Y.; software and validation, H.Y. (Haodong Yan) and X.Z.; writing—original draft preparation, H.Y. (Hua Ye) All authors have read and agreed to the published version of the manuscript.

**Funding:** National Natural Science Foundation of China (Grant No. 51909047); National Key R&D Program of China (Grant 2019YFB1705302); Shandong Provincial Key Research and Development Plan (Grant No. 2021CXGC010702 and No. 2019GHZ011).

**Institutional Review Board Statement:** Not applicable.

**Informed Consent Statement:** Not applicable.

**Data Availability Statement:** Not applicable.

**Acknowledgments:** Not applicable.

**Conflicts of Interest:** The authors declare no conflict of interest.

**Appendix A**

The steps for achieving the UKF are as follows.

(1) Initialize the mean weights of the state variables $\omega_m$ and the error covariance weights $\omega_c$:

$$\omega_m = \frac{\lambda}{n+\lambda} \tag{A1}$$

$$\begin{cases} \omega_{c,0} = \frac{\lambda}{n+\lambda} + 1 - \alpha^2 + \beta \\ \omega_{c,k} = \frac{1}{2(n+\lambda)}, k \neq 0 \end{cases} \tag{A2}$$

where $\omega_m$ and $\omega_c$ are, respectively, the weight of the mean value of the state variable and the weight of the error covariance of the state variable; $\lambda = \alpha^2(n_{dim} + \kappa) - n_{dim}$ represents the distance between sigma point and mean value; $\beta$ represents prior knowledge of mean distribution; $\kappa$ similar to $\alpha$, is a secondary scaling parameter; $n_{dim}$ represents the dimensions of the sampling data;

(2) Calculate the sigma points $\widetilde{x}$ and $\widetilde{y}$:

$$\begin{cases} \widetilde{x}_{0,k-1} = \mu_{k-1} \\ \widetilde{x}_{i,k-1} = \mu_{k-1} + \sqrt{(n_{dim} + \lambda)P_{xx,k-1}}, i = 1, 2, \ldots, n_{dim} \\ \widetilde{x}_{i,k-1} = \mu_{k-1} - \sqrt{(n_{dim} + \lambda)P_{xx,k-1}}, i = n_{dim} + 1, n_{dim} + 2, \ldots, 2n_{dim} \end{cases} \tag{A3}$$

$$\begin{cases} \widetilde{x}_k = f(\widetilde{x}_{k-1}) \\ \widetilde{y}_k = g(\widetilde{x}_k) \end{cases} \tag{A4}$$

where $\widetilde{x}_i$ and $P_{xx}$ respectively represent the state variables corresponding to the ith sigma point and their error covariance; $\mu$ represents the mean value of the state variable; $\widetilde{y}$ represents the output variable predicted by sigma points;

(3) Predict the state variables $\hat{x}$ and error covariance $\hat{P}_{xx}$:

$$\hat{x}_k = \sum_{i=0}^{2n_{dim}} \omega_{m,i} \widetilde{x}_{i,k} \tag{A5}$$

$$\hat{P}_{xx,k} = \sum_{i=0}^{2n_{dim}} \omega_{c,i} (\widetilde{x}_{i,k} - x_k)(\widetilde{x}_{i,k} - x_k)^T + \hat{Q}_k \tag{A6}$$

where $\hat{x}$ and $\hat{P}_{xx}$ respectively represent the state variables and their error covariance calculated from the unscented transformation; $\hat{Q}_k$ represents the covariance of systematic error at time $k$;

(4) Predict the output variables $\hat{y}$ and error covariance $\hat{P}_{yy}$:

$$\hat{y}_k = \sum_{i=0}^{2n_{dim}} \omega_{m,i} \widetilde{y}_{i,k} \tag{A7}$$

$$\hat{P}_{yy,k} = \sum_{i=0}^{2n_{dim}} \omega_{c,i} \left(\widetilde{y}_{i,k} - y_k\right)\left(\widetilde{y}_{i,k} - y_k\right)^T + \hat{R}_k \tag{A8}$$

where $\hat{y}$ and $\hat{P}_{yy}$ respectively represent the output variables and their error covariance calculated from the unscented transformation; $\hat{R}_k$ represents the covariance of measurement error at time $k$;

(5) Calculate the Kalman gain $K$ and residuals $e$:

$$P_{xy,k} = \sum_{i=0}^{2n_{dim}} \omega_{c,i} (\widetilde{x}_{i,k} - \hat{x}_k)\left(\widetilde{y}_{i,k} - \hat{y}_k\right) \tag{A9}$$

$$K_k = \hat{P}_{xy,k} \hat{P}_{yy,k}^{-1} \tag{A10}$$

$$e_k = y_k - \hat{y}_k \tag{A11}$$

where $\hat{P}_{xy}$ represents the error covariance of the state variable and the output variable; $K$ represents Kalman gain; $e$ represents residual.

(6) Update the state variables $x$ and error covariance $P_{xx}$:

$$x_k = \hat{x}_k + K_k e_k \tag{A12}$$

$$P_{xx,k} = \hat{P}_{xx,k} - K_k \hat{P}_{yy,k} K_k \tag{A13}$$

where $P_{xx}$ represents the error covariance of the updated state variable.

Through the prediction and correction process, the unscented Kalman filter can achieve signal denoising, optimal estimation, and other functions. The effect of Kalman filtering depends on the state transformation equation and measurement transformation equation of the system. Therefore, different results can be obtained through different transformation equations. The smaller the residual error calculated according to Equation (A11), the closer the corresponding system dynamic characteristics are to the current time.

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
