# Peer review of "A Multi-Model Diagnosis Method for Slowly Varying Faults of Plunger Pump"

_jmse, doi:10.3390/jmse10121968_

Round 1

Reviewer 1 Report

The authors propose a multiple-model diagnosis method for slowly varying faults of marine equipment. Although I believe that the subject of this work is worthy of investigation, I believe that the work needs to respond to some concerns that I point out below:

The literature review presented by the authors is rather poor. Only a dozen other works are cited throughout the text. Is the amount of work that deals with this topic really that small?

Chapter 2, SVM - MM Algorithm Framework is really hard to follow. I believe that the authors can present a very detailed flowchart of all the steps for the implementation of the proposed method. In this way, it is easier for the reader to understand the authors' ideas and makes the work reproducible.

In item 2.5, the authors say that pthreshold = 0.95. Why was this specific value chosen?

In chapter 3, the authors present the mathematical model of the plunger pump, the theoretical example chosen for the application of the proposed method, and, in item 3.2, they present which faults of the plunger pump were considered. But I didn't understand exactly how these faults were simulated. For example, according to Table 2, when there is a shaft displacement, a swash plate inclination of γ - 2 occurs, but is this inclination sudden or does it increase with time until it reaches a threshold value? Authors need to better explain how the faults were simulated and add graphs that show the behavior of parameters over time, which would help readers better understand what type of fault is being diagnosed.

Still on item 3.2, for fault diagnosis to occur, parameters must first be observed by sensors and constantly monitored. Are the parameters listed in Table 2 monitorable? What types of sensors could be used to monitor these types of parameters?

In item 4.1, the authors say: "to simulate a real measurement situation, we randomly add 1% disturbance to data". What kind of disturbance was added? And why 1%?

The authors use a theoretical example to prove that the proposed method, which combines MM and SVM, is better than the traditional MM. But in order for such a claim to actually be proved, they need to show that the methods have been applied to a dataset that can in fact be used for that purpose. The way the authors present the fault simulation did not convince me that, in a real scenario, SVM-MM is better than MM. I believe that there are two possible alternatives that the authors can consider to show that the proposed method is in fact better: either they explain better how the failures were simulated; or they use a dataset already used in other works, such as CMAPSS (Commercial Modular Aero-Propulsion System Simulation).

Reviewer 2 Report

This paper introduces a support vector machine based algorithm for condition monitoring of marine equipment. Indeed, implementation of artificial intelligence to diagnosis is an actual and extremely important topic nowadays. There are the comments to be considered:

1. The introduction requires more relevant references to introduce similar works done over the last years. Please, add also some paragraph about common fault in marine equipment and add respective references.

2. What is the main novelty of this paper?

3. The graphs must be more informative. Besides, please check the captions of the pictures as well as add clarifications to them.

4. Please, follow the template.

5. Do you compare the simulation results with the practical data of the real fault in equipment?

6. The title says that there is a diagnosis of “varying faults of marine equipment”. However, there were presented only plunger pump faults. Will you consider other fault types?

Round 2

Reviewer 1 Report

I believe the authors have responded appropriately to all of my concerns.